geophysics

small-scale mining, resistivity, induced polarization, mine planning, mining sustainability

**Author for correspondence:**
Ricardo Tichauer
e-mail: ricardo.tichauer@gmail.com

# The role of geophysics in enhancing mine planning decision-making in small-scale mining

## Ricardo Tichauer, Antonio Carlos Martins, Ranyere Sousa Silva and Giorgio De Tomi

Department of Mining and Petroleum Engineering, USP - Universidade de Sao Paulo, Sao Paulo, Brazil

(iD) RT, 0000-0001-8350-2126

Small-scale mining usually operates under high geological uncertainty conditions. This turns mine planning into a complex and sometimes inaccurate task, resulting in low productivity and substantial variability in the quantity and quality of the mineral products. This research demonstrates how the application of a novel methodology that relies on traditional and low-cost geophysical methods can contribute to mine planning in small-scale mining. A combination of resistivity and induced polarization methods is applied to enhance mine planning decision-making in three small-scale mining operations. This approach allows for the acquisition of new data regarding local geological settings, supporting geological modelling and enhancing decision-making processes for mine planning in a timely and low-cost fashion. The results indicate time savings of up to 77% and cost reductions of up to 94% as compared with conventional methods, contributing to more effective mine planning and, ultimately, improving sustainability in small-scale mining.

## 1. Introduction

Small-scale mining has played an increasingly important role for communities that depend on this activity worldwide. Buxton [1] estimates that small-scale mining employs 20 to 30 million people in over 80 countries, and that this sector is responsible for 15 to 20% of the global production of minerals and metals. Villegas *et al.* [2] report that small-scale mining supplies 80% of gemstones, 25 to 30% of tin, 15 to 20% of diamonds and 10% of gold in the world. According to Ericsson [3], the depletion of large deposits with high grades is driving increased investment in medium- and small-scale mineral production projects. However, there are many

challenges that keep small-scale mining from fulfilling its vocation and potential in promoting socioeconomic development for communities in urban peripheral areas and in rural regions, including remote locations that predominantly depend on this activity. Gamarra Chilmaza [4] points out that small-scale mining is often an informal activity, with low adherence to legal requirements. Leite *et al.* [5] refer to small-scale mining as an activity that often operates with simple, often rudimentary equipment and tools.

One of the main risks that small-scale mining face is operating under insufficient geological information for efficient mine planning. Hruschka & Echavarria [6] explain that the search for geological knowledge presents numerous challenges for small-scale mining that is often neglected because of insufficient capital. Hentschel *et al.* [7] also mention that a shortage of investment capital for exploration is one of the main causes of inefficiency in small-scale mines. Tichauer [8] reports that, because of insufficient funding for mineral exploration, small-scale mining frequently operates at a high level of geological uncertainty. However, geological knowledge is a major factor for success in mining. Abichequer *et al.* [9] state that geological uncertainty is one of the main causes of failure in mining. For Godoy [10], the most important risk in mining is that associated with geological uncertainty.

Many researchers have demonstrated that geological information is a critical driver of efficiency in mine planning [11–16]. This study aims to answer the following research questions: how can geological uncertainty be reduced to a level that allows for efficient mine planning in the short-, medium- and long-term horizons in small-scale mining? What methods can be used to address this challenge? Traditional and low-cost geophysical methods such as electrical resistivity and induced polarization (IP) can be the necessary tools to respond these questions.

# 2. Material and methods

Geophysics has been traditionally employed in exploration for identification and delimitation of mineral deposits and has played an important role in reducing geological uncertainty in mining. Haile & Atsbaha [17] comment that geophysical techniques are routinely used as part of geological investigations to map subsurface geological structures. According to Frasheri *et al.* [18], the most common geophysical methods employed in mineral exploration are electrical, electromagnetic, gravimetric, magnetic and seismic. Shallow geophysics, for example, is used for the investigation of geological structures with small dimensions at the top of the crust.

Currently, shallow geophysics is frequently used in mining [19,20]. The methods of resistivity and IP have been often applied to deposits of sulfides, metals and graphite [21]. Moreira *et al.* [22] show how the application of resistivity and IP supplied data for modelling of a gold deposit. Coelho *et al.* [23] aim to evaluate the potential of resistivity tomography as a prospecting tool for supergene ore. Martins *et al.* [24] demonstrate how resistivity and IP can assist on morphological modelling in a limestone mine. Martins *et al.* [25] contribute to mineral exploration in small-scale mining by showing how resistivity and IP can provide important information for geological modelling quickly at a low cost and can reduce, by 30–50%, the amount of drill holes that do not find ore. Moreira *et al.* [26] also recommends resistivity tomography for mineral exploration because of the quickness of the procedures and the reduction of project costs. Therefore, the application of geophysical methods during all phases of a small-scale mining project can reduce geological uncertainty by assisting the identification and delineation of mineral deposits and by delivering important information for decision-making in mine planning.

## 2.1. Proposed methodology

The proposed methodology is illustrated in figure 1. It relies on traditional, low-cost geophysics methods to lower geological uncertainty of small mineral deposits to allow for efficient strategic (long-term), tactical (medium-term) and operational (short-term) mine planning horizons in small-scale mining. The first step is to define the mine-planning horizon of the project. Once decided if it is a strategical, tactical or operational horizon, an evaluation matrix is applied to indicate the level of geological uncertainty in the deposit before the application of geophysics. The next step is the selection of the appropriate geophysics method according to the chosen mine-planning horizon. The resulting information from the geophysical study is processed to generate a geological model and a conceptual pit design for that deposit. The assessment of how the selected planning horizon can be developed results on the conceptual production plan for the project. Then, the evaluation matrix is re-applied to assess the reduction of geological uncertainty associated with the project. At the end, a decision is made on

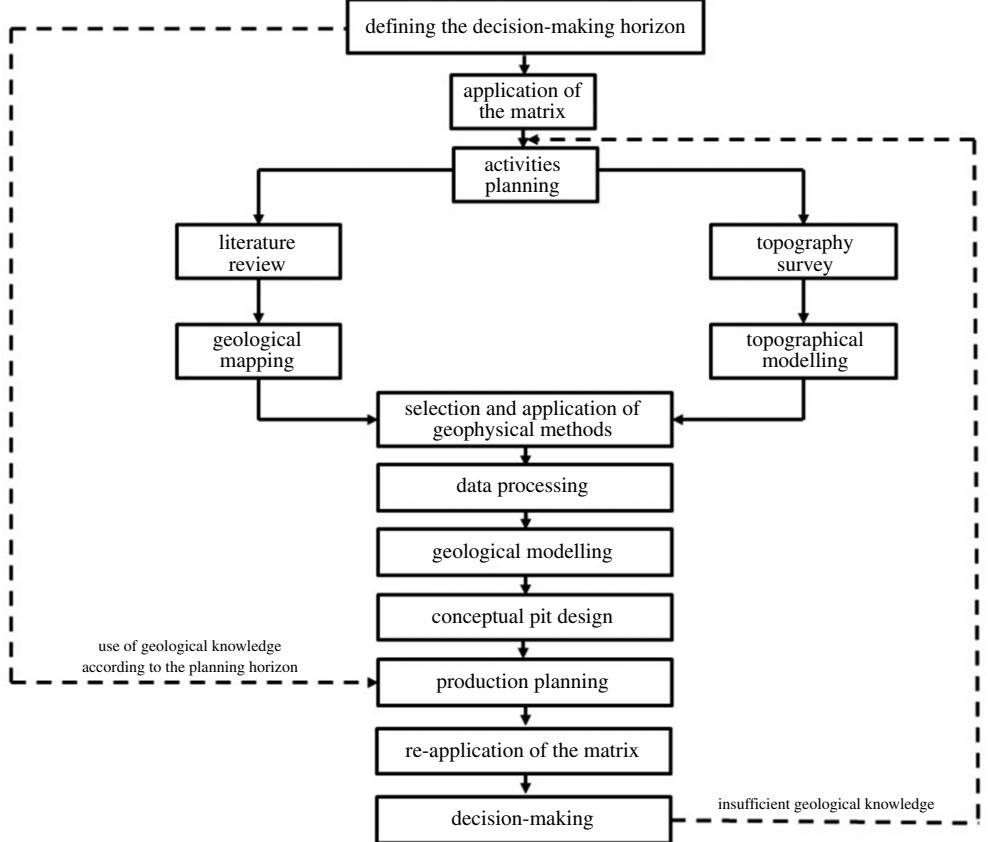

**Figure 1.** Proposed methodology.

confirming the conceptual production plan for the proposed horizon or, in case of insufficient information, further geophysical studies can be carried out until enough data is obtained to support a solid production plan. This methodology relies on the application of the two geophysical methods described below.

## 2.2. Geophysical methods

The specific geophysical methods employed in the proposed methodology are those of resistivity and IP. Resistivity data is obtained through two complementary techniques, vertical electrical sounding (VES) with a Schlumberger array and electrical profiling (EP), with a dipole–dipole array. The method of resistivity can be used for identification and delimitation of depth and width of the subsurface layers. IP data complements and confirms resistivity information and is also gathered by EP surveys [27–29].

VES consists of introducing an artificial current on the ground through two electrodes, A and B. The potential generated in other two electrodes near the current flow, M and N, is used for the calculation of the apparent electrical resistivity in the subsurface. Increasing the distance between the two current electrodes allows the current to reach deeper layers, as illustrated in figure 2 [30]. The successive results indicate the depth and thickness of geological layers based on variations of resistivity. VES is especially useful for plain areas. According to Sahbi *et al.* [31], when VES is applied along irregular terrains, topographical effects can influence the values of apparent electrical resistivity and lead to erroneous interpretation.

Electrical profiling is performed along survey lines in the terrain, resulting in a two-dimensional profile for each line. The data obtained relate to each other through the investigation of the layers depth and thickness. While VES surveys generate vertical and deeper data, EP information and interpretation are displayed in sections showing subsurface layers at lower depths. Figure 3 shows how the electrodes are disposed and transmit current flows in EP with the dipole–dipole array [32].

IP is an electrical phenomenon stimulated by electric current. The application of an electric current to the surface results in a V primary difference of potential, which, in some situations, provokes polarization

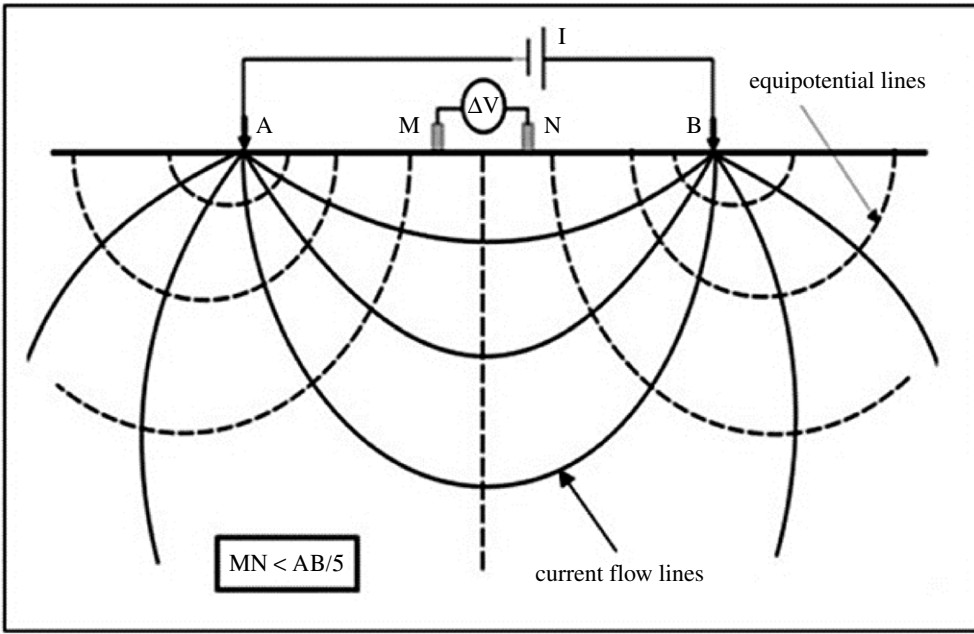

**Figure 2.** Disposition of electrodes for vertical electrical sounding (VES) (modified from [30]).

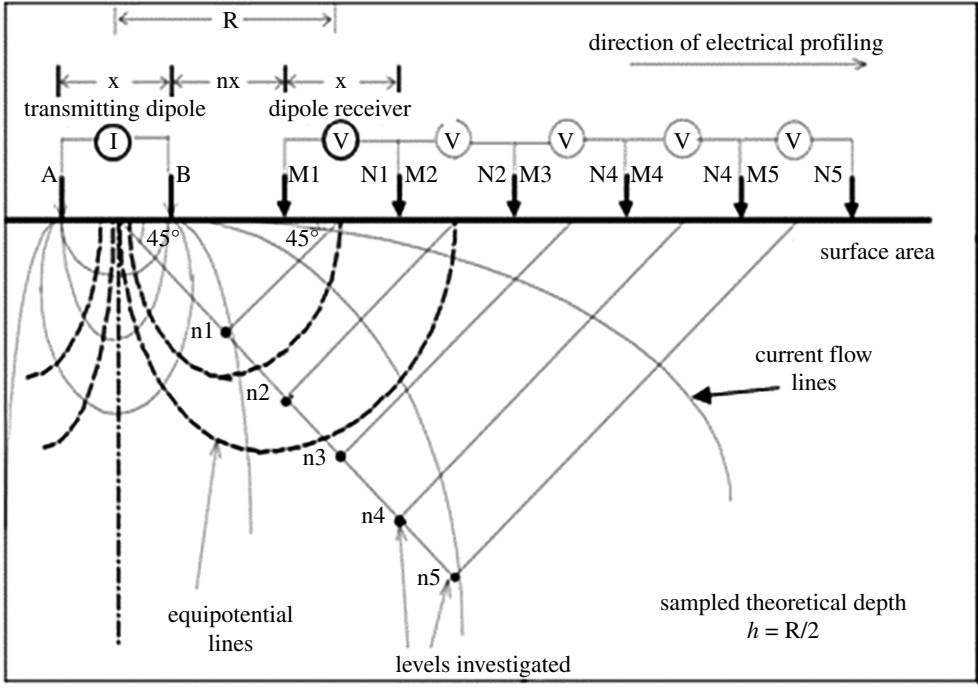

**Figure 3.** Disposition of electrodes for electrical profiling (EP) with dipole–dipole array (modified from [32]).

of materials in the subsurface. This capacity of polarization constitutes the IP susceptibility of the subsurface materials. The measured parameter is called chargeability.

Resistivity and IP are normally measured together, because an electric current is used to generate a primary difference of potential. The equipment reads the resistivity of a current flow for a period that usually ranges from 1 to 4 s, using the primary V. Traditionally, IP is used in exploration for disseminated sulfides. It is rarely used for oxides and hydroxides, such as primary and secondary manganese ore [33].

The equipment employed for the application of these methods includes an Iris Syscal Transmitter and an Iris Elrec Pro Receiver. The components for measuring the difference of electrical potential are non-polarized electrodes with copper sulfide solution. The parameters of acquisition are integration time of 2 s; delay time of 0.16 s and minimal stacking of 10 measurements. The stacking extends up to

pre-assigned weights

score 1, 2, 3, 4 or 5 for level of implementation

| project xyz | cost | effort | benefits | CEB | IMPL | index |
|---|---|---|---|---|---|---|
| | maxC = 5 | maxE = 5 | maxB = 5 | max = 625 | >80% = 5 | (2–10) |
| guidelines: | minC = 1 | minE = 1 | minB= 1 | min = 5 | <20% = 1 | |
| 1 exploration planned and conducted professionally. | 3 | 3 | 5 | 45 | 3 | 0.6 |
| 2 geological assumptions based on scientific approach. | 2 | 2 | 5 | 20 | 3 | 0.3 |
| 3 quality control applied to data acquisition. | 3 | 3 | 5 | 45 | 2 | 0.4 |
| 4 proper registration of methods and data. | 3 | 2 | 5 | 30 | 2 | 0.3 |
| 5 sampling performed with scientific basis. | 4 | 4 | 5 | 80 | 2 | 0.7 |
| 6 suitable drilling methods. | 5 | 5 | 5 | 125 | 2 | 1.1 |
| 7 samples managed with propersecurity. | 2 | 2 | 4 | 16 | 3 | 0.2 |
| 8 adequate preparation of samples. | 2 | 3 | 4 | 24 | 3 | 0.3 |
| 9 analysis of samples carried out by reputable laboratory. | 2 | 2 | 4 | 16 | 5 | 0.4 |
| 10 works carried out in compliance with legislation. | 3 | 4 | 4 | 48 | 4 | 0.9 |
| | | | | | | 5.1 |

the Tichauer-De Tomi index

**Figure 4.** Evaluation matrix showing a hypothetical project with an index of 5.1 (modified from [34]).

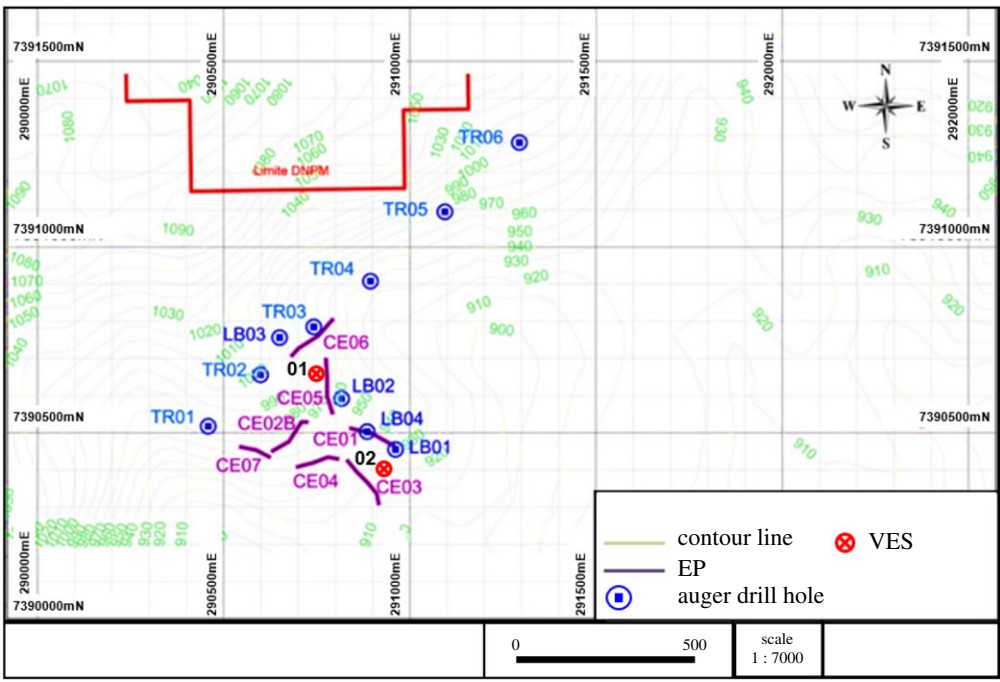

**Figure 5.** Location of auger, vertical electrical sounding (VES) and electrical profiling (EP) surveys (modified from [35]).

**Table 1.** Exploration techniques applied to the three small-scale mining projects (LTP, long-term plan; MTP, medium-term plan; STP, short-term plan).

| | techniques | | |
|---|---|---|---|
| project | auger | VES | EP |
| gold (LTP) | x | x | x |
| manganese (MTP) | | x | x |
| limestone (STP) | | x | |

20 measurements if the standard deviation among consecutive measurements is above 3%. Processing is carried out with software Iris Prosys II and excludes measurements with standard deviation above 3%. The VES data is interpreted with the inversion software IX1D (Interpex, 2013). The interpretation of the

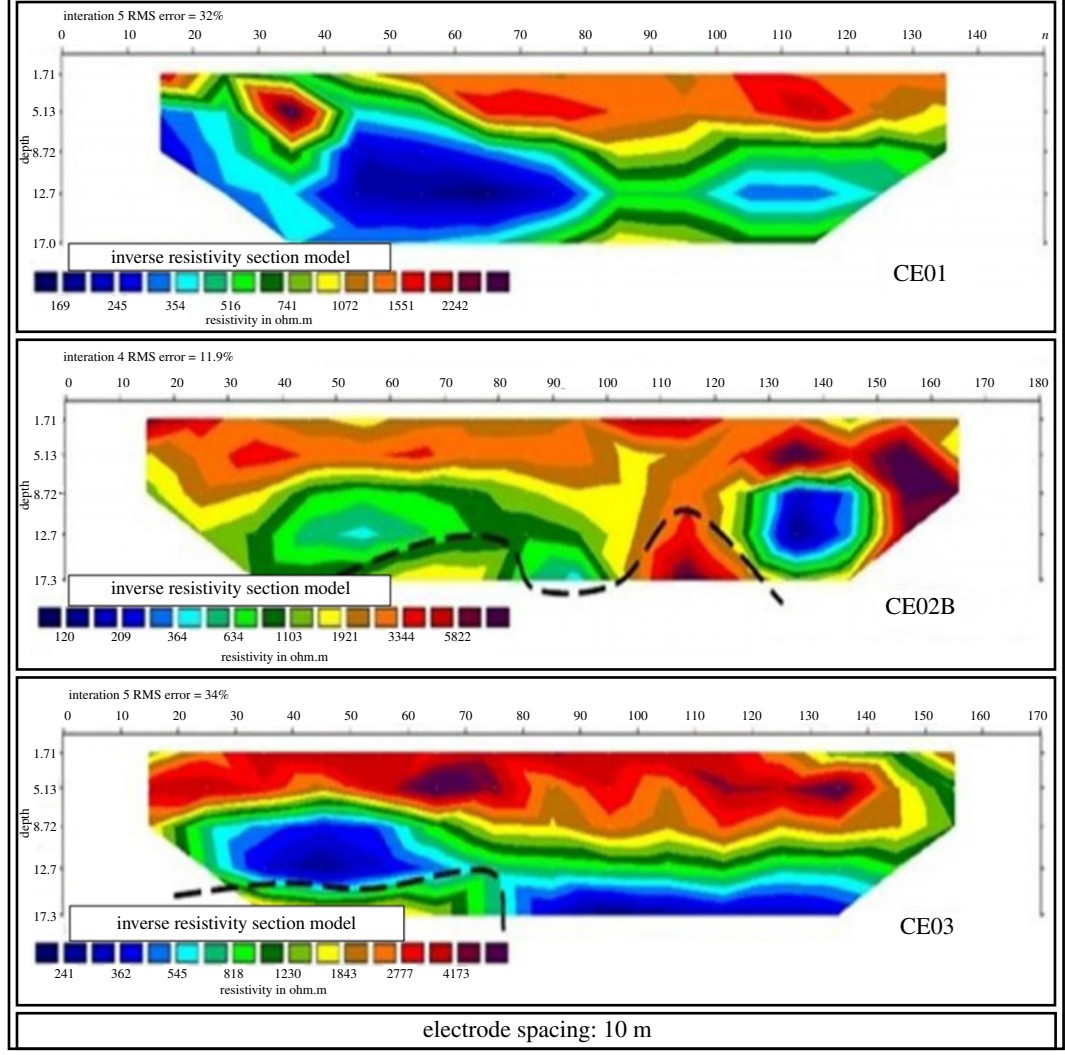

**Figure 6.** Electrical profiling sections CE01, CE02B and CE03 (modified from [35]).

results relies on the method of successive approximations or inversion by chain regression. The apparent resistivity curve is based on the parameters of resistivity, chargeability and the widths of the initial mode. By adjusting this curve to the field curve, the initial parameters are modified automatically so to achieve the best fit of the initial model. The data obtained from EP are interpreted with the software RES2DINV (Geotomo, 2009) through the process of inversion by least-squares, which transforms field data in modelled resistivity profiles that show values of the actual resistivity and depth.

## 2.3. Evaluation matrix

The matrix proposed by Tichauer & De Tomi [34] measures the level of compliance of mineral exploration programmes with industry best practices. This matrix is composed of 10 guidelines, based on the directives for mineral exploration established by the Canadian National Instrument 43-101 (NI 43-101). The guidelines are weighed based on the specific impact of cost, effort and benefits associated with the implementation of the guideline.

A surveyor must assess each guideline and assign a score for the level of implementation for each one of the 10 guidelines. Scores are integer numbers from 1 to 5 and measure the quality and broadness of implementation for each guideline. The assignment of scores for each guideline results in the matrix index.

The matrix index ranges from 2.0 to 10.0. An index of 2.0 means that the average level of implementation of best practices lies between 0% and 20%, and represents knowledge limited to basic local geological understanding, the existence of outcrops or unreliable exploration information. An index of 10.0 means that the average level of implementation of best practices is between 80% and

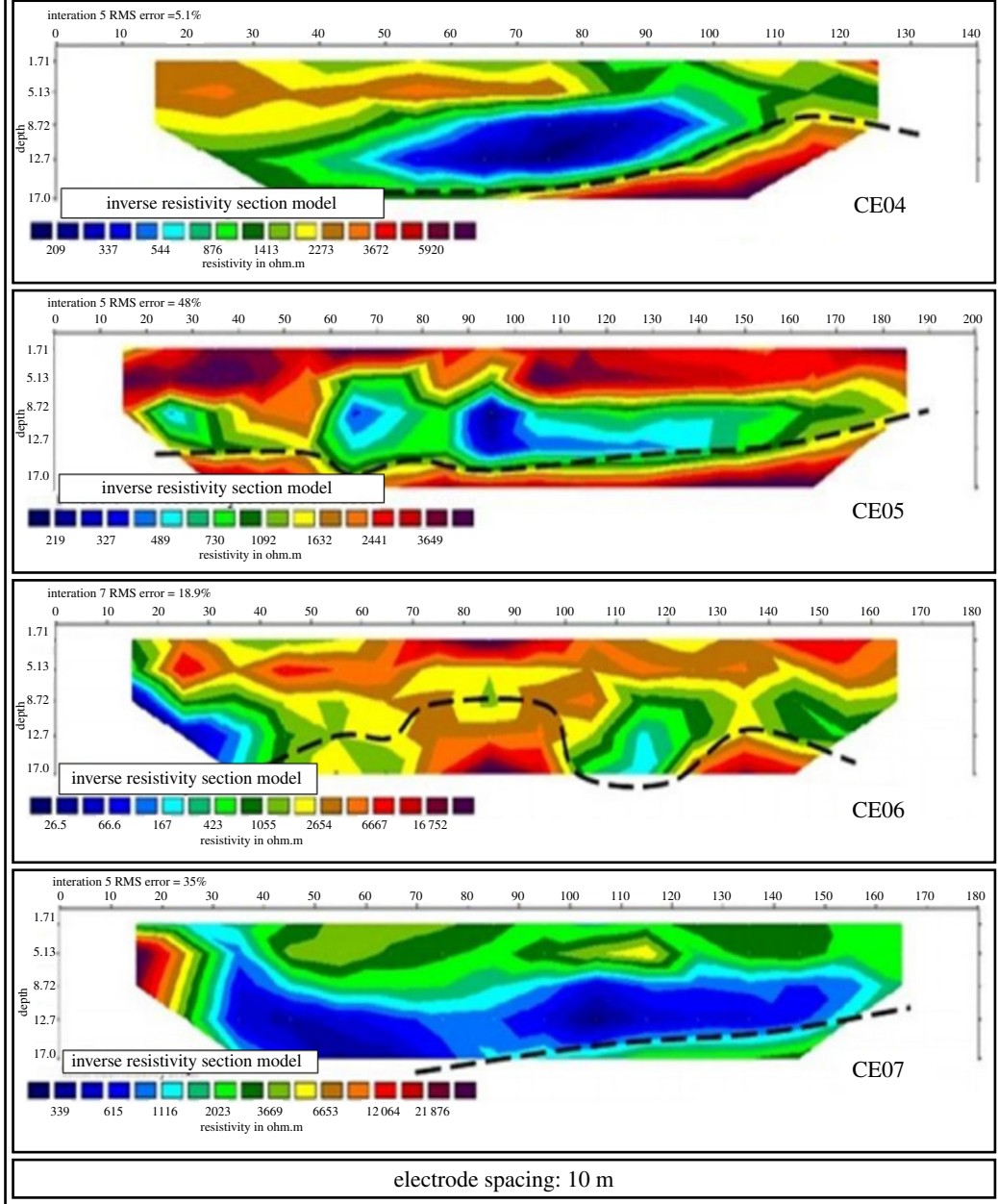

**Figure 7.** Electrical profiling sections CE04, CE05, CE06 and CE07 (modified from [35]).

100% and represents a satisfactory level of compliance with internationally accepted exploration standards, providing reliable results to support a solid mining plan.

Figure 4 shows the weights for cost, effort and benefits for each guideline in the matrix and displays scores assigned for a hypothetical exploration programme. The index of 5.1 indicates that the average level of implementation lies between 31% and 51%.

## 3. Results

The proposed methodology has been applied to three mining projects at different stages of development: a gold, a manganese, and a limestone small-scale mining projects in the state of São Paulo, Brazil. Each one of these projects was in a different horizon of mine planning decision-making: the gold project was in the initial stages of assessment and geophysics was applied to support strategic decision-making for a long-term mining plan; the manganese deposit had already been partially mined out and geophysics was carried on for tactical decision-making for a medium-term mine plan;

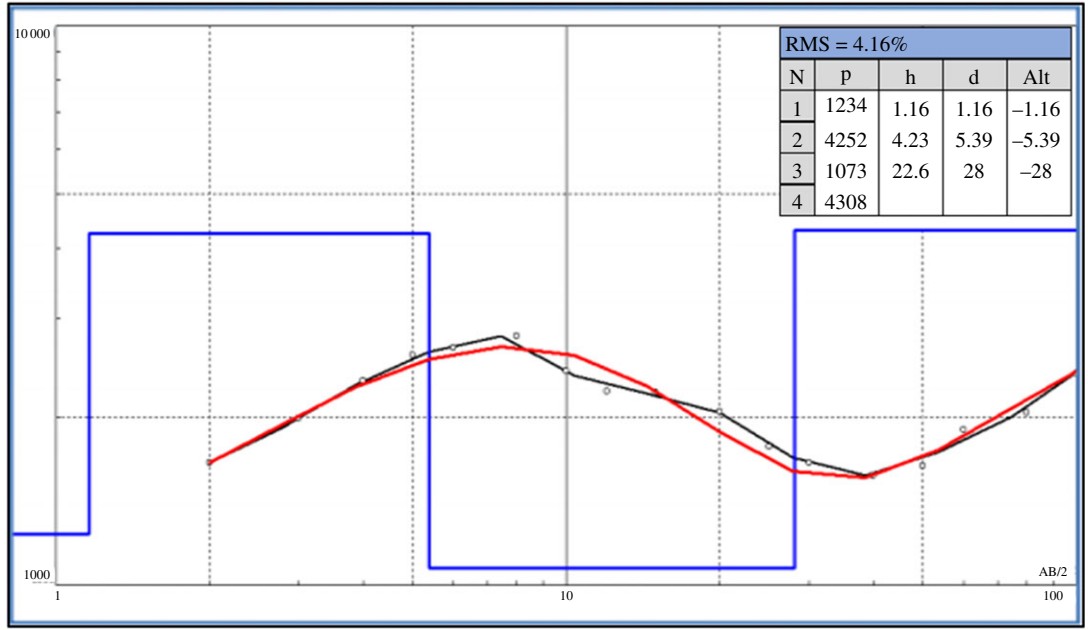

**Figure 8.** VES01 survey (modified from [35]).

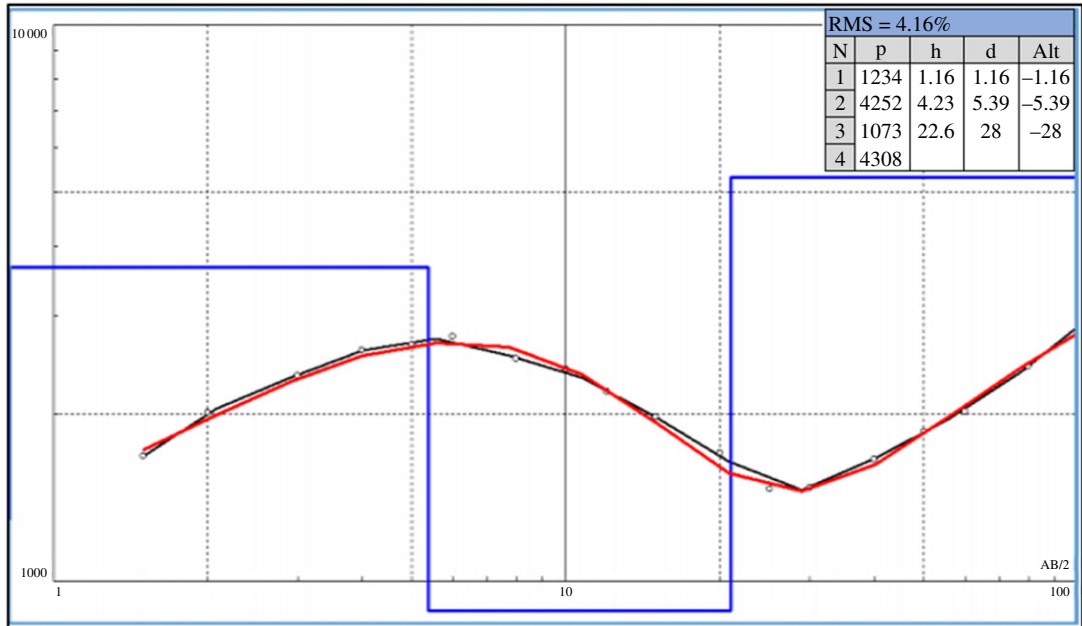

**Figure 9.** VES02 survey (modified from [35]).

and the limestone mine has been operating for a number of years and geophysics was conducted for operational decision-making for a short-term mine plan. Table 1 indicates the data acquisition methods applied to each one of these projects.

## 3.1. Long-term decision-making: gold project

Initial drilling in the target area consisted of eight 4 m auger drill holes. Samples were collected from the intervals between 0 and 2 m and between 2 and 4 m of depth. The analysis of the samples showed grades up to 90 ppb. However, information from the literature, local geological mapping and additional drilling indicated that higher grades may be found in deeper levels of the local structure. As the auger drill holes did not reach the top of the bedrock, geophysical methods were employed to complement the understanding of the local geology and the gold occurrence. The geophysics works were planned and executed as follows:

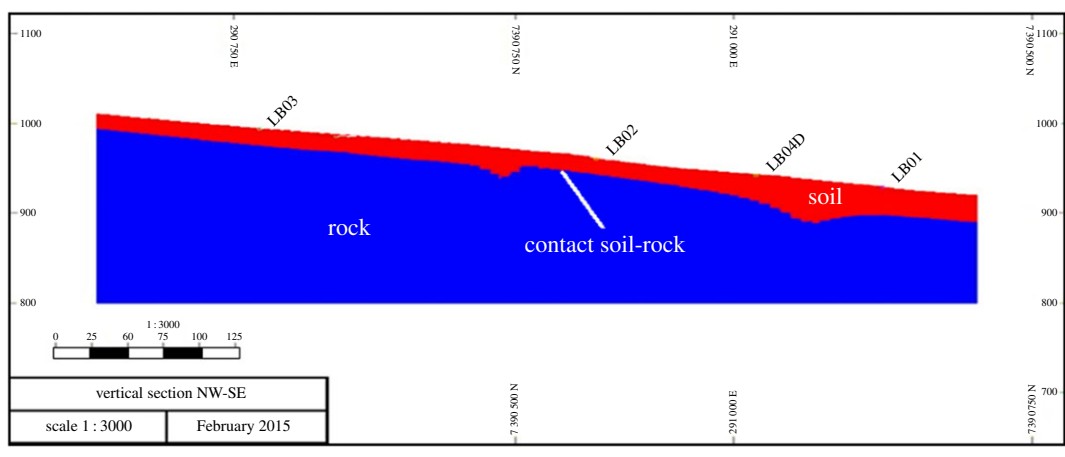

**Figure 10.** Profile resulting from geophysical information (modified from [35]).

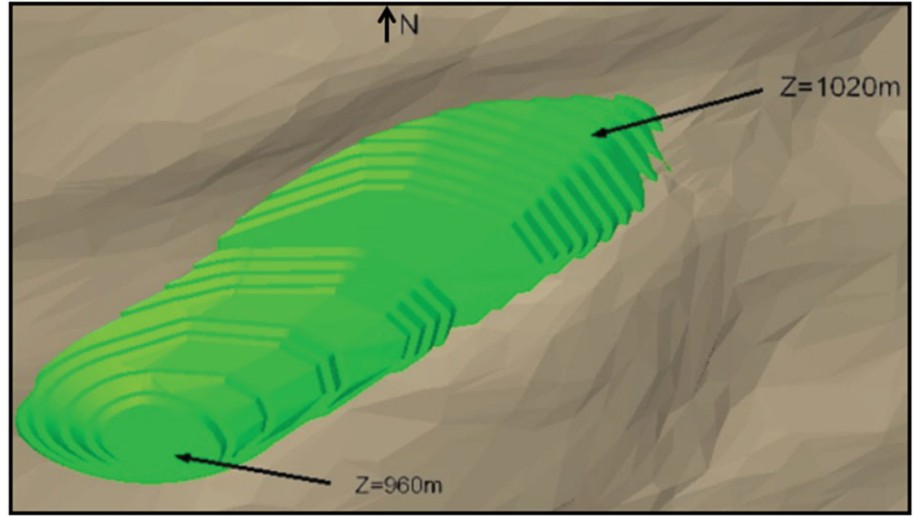

**Figure 11.** Conceptual pit design for the gold deposit (modified from [35]).

**Table 2.** Results from VES01 and VES02 surveys (modified from [35]).

| VES | from (m) | to (m) | stratigraphy |
|-----|----------|--------|--------------|
| 01 | 0.0 | 1.2 | soil |
| | 1.2 | 5.4 | altered rock |
| | 5.4 | 28.0 | altered rock—high moisture |
| | 28.0 | — | rock |
| 02 | 0.0 | 0.8 | soil |
| | 0.8 | 5.4 | altered rock |
| | 5.4 | 21.0 | altered rock—high moisture |
| | 21.0 | — | rock |

  (i) seven EP surveys ranging from 140 m to 200 m long; and
 (ii) two VES surveys.

Figure 5 shows the property and the location of the auger drill holes, the EP activities and the VES surveys carried out. The profile of the EP surveys CE01, CE02B and CE03 are presented in figure 6. The profile of the EP surveys CE04, CE05, CE06 and CE07 are shown in figure 7. The diagram for the

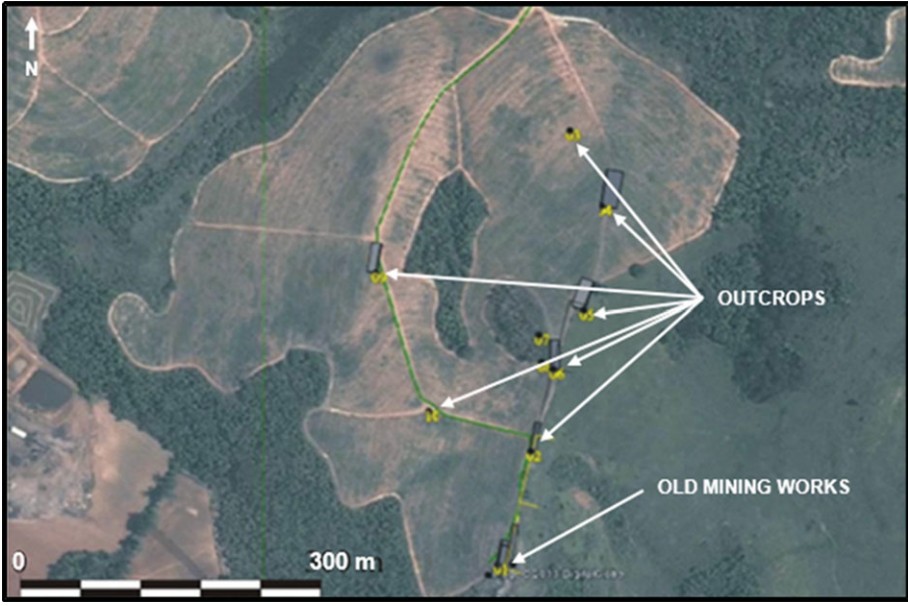

**Figure 12.** Aerial image of the surveying target with manganese-rich outcrops (modified from [35]).

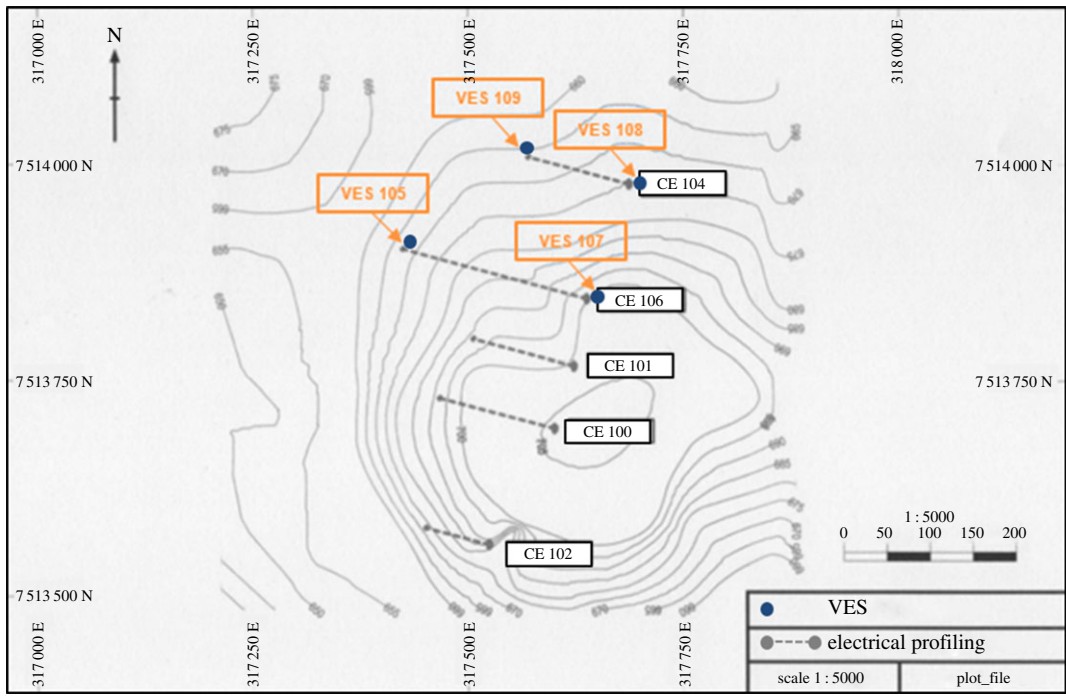

**Figure 13.** Location of vertical electrical sounding (VES) and electrical profiling (EP) surveys (modified from [24,25]).

VES01 survey is displayed in figure 8, and for the VES02, in figure 9. Table 2 shows the soil and rock layers identified through the VES results. Geophysics, combined with the auger drilling campaign, confirmed the existence of a gold mineralized soil layer, as shown by the profile in figure 10 [35].

The goal was the assessment of the preliminary potential of the property for gold production. Therefore, the information was gathered and a conceptual pit was modelled for a strategic decision about making an investment to move forward and follow all the necessary steps to carry out further studies in the area. The information obtained through the geophysical survey was appropriate for the elaboration of a conceptual pit design, as shown in figure 11. The ultimate conceptual pit designed contributed to strategic decision-making based on mining costs and gold production potential in the long-term horizon of the project.

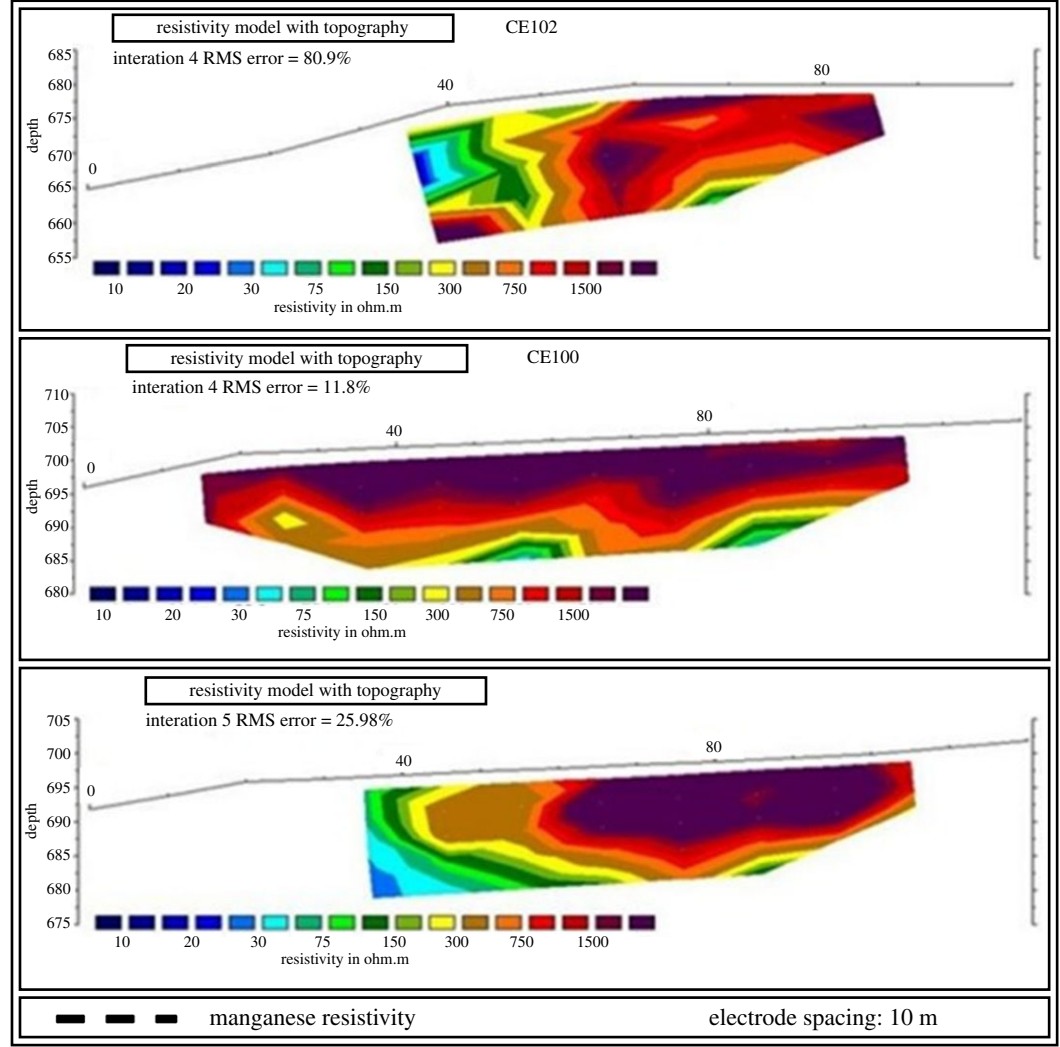

**Figure 14.** Electrical profiling sections CE102, CE100 and CE101 (modified from [35]).

## 3.2. Medium-term decision-making: manganese deposit

The works were carried out in an area where there were outcrops of manganese-rich layers and old mining works. Figure 12 shows the map where manganese-rich outcrops were identified. The fieldwork carried out included topographic surveying, geological mapping and geophysical surveying. The data interpretation generated a preliminary geological model and a drilling plan. VES with the Schlumberger array was chosen for its effectiveness in detecting low depth subsurface layers with low sensitivity to lateral variations of resistivity. In EP, the dipole–dipole array was employed as it also obtains detailed and precise information regarding the thickness and depth of geological layers. The geophysics works were planned and executed as follows:

 (i) four long EP surveys spaced at 150 m from each other (CE104, CE100 and CE102 were 120 m long, and CE106, 240 m long);

 (ii) one 120 m long EP survey (CE101) spaced at 75  m from CE106 and CE100; and

 (iii) four VES surveys.

The five EP surveys were carried out in the northwest-southeast direction, transverse to the deposit axis, according to figure 13 and resulted in the sections presented in figure 14 (CE100, CE101 and CE102) and in figure 15 (CE104 and CE106). The dashed lines in the sections separate the manganese bodies with low resisitivity (colours yellow, green and blue) from the other layers (colours brown, orange, red and black). The data obtained through VES and IP activities indicate that the mineralized layer can exceed 10 m thick and that the portions of the deposit can be found over 40 m of depth, and that a significant part of the

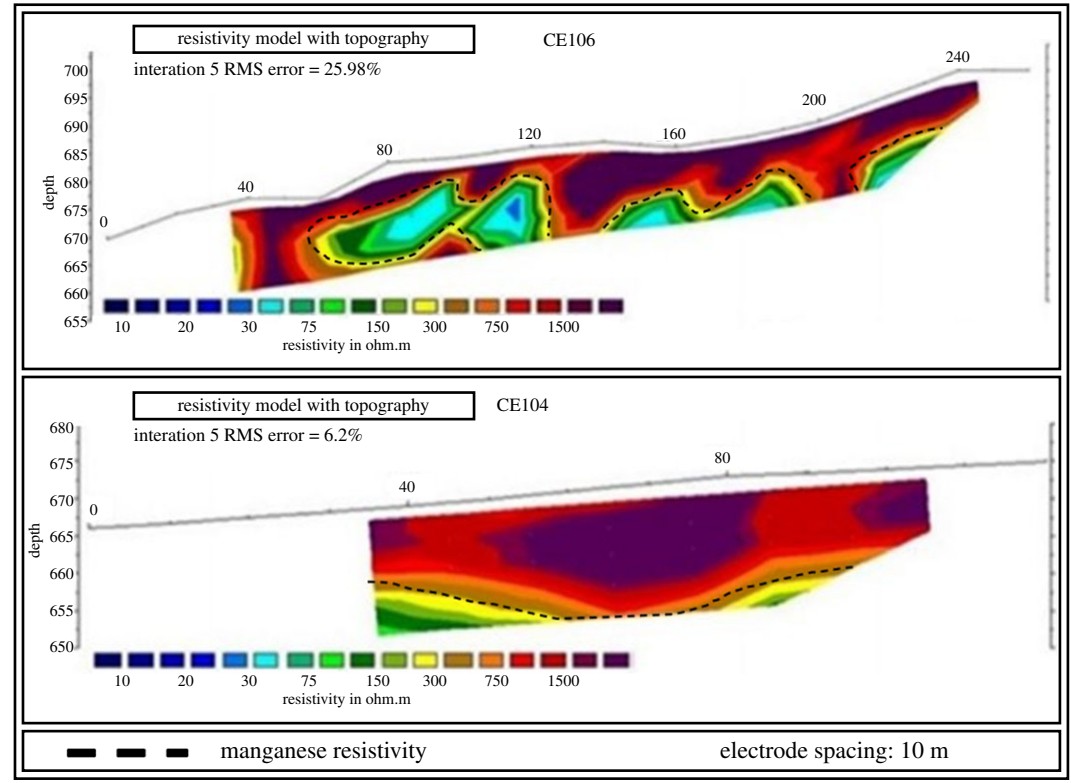

**Figure 15.** Electrical profiling sections CE106 and CE104 (modified from [35]).

**Table 3.** Interpretation of VES information (modified from [24,25]).

| VES | from (m) | to (m) | stratigraphy |
|-----|----------|--------|--------------|
| 105 | 0.0 | 1.1 | soil |
|     | 1.1 | 35.0 | altered rock |
|     | 35.0 | — | manganese |
| 107 | 0.0 | 9.8 | soil |
|     | 9.8 | 28.0 | altered rock |
|     | 28.0 | — | manganese |
| 108 | 0.0 | 13.5 | soil |
|     | 13.5 | 18.6 | altered rock |
|     | 18.6 | 29.0 | manganese |
| 109 | 0.0 | 6.6 | soil |
|     | 6.6 | 42.9 | altered rock |
|     | 42.9 | 55.0 | manganese |
|     | 55.0 | — | rock |

manganese ore body can be found between 18.6 and 42.9 m of depth. Table 3 shows the interpretation of VES information. Because the IP method is usually employed for exploration of disseminated sulfides and the manganese ore in the property presents a small quantity of sulfides, the high values of chargeability confirmed the position of the mineralized bodies [25].

This property had been mined before and already had all mining licenses and processing equipment necessary to resume operations. The combination of geophysical data contributed to the elaboration of a preliminary model of the manganese deposit at the target area, as seen in figure 16, and the conceptual

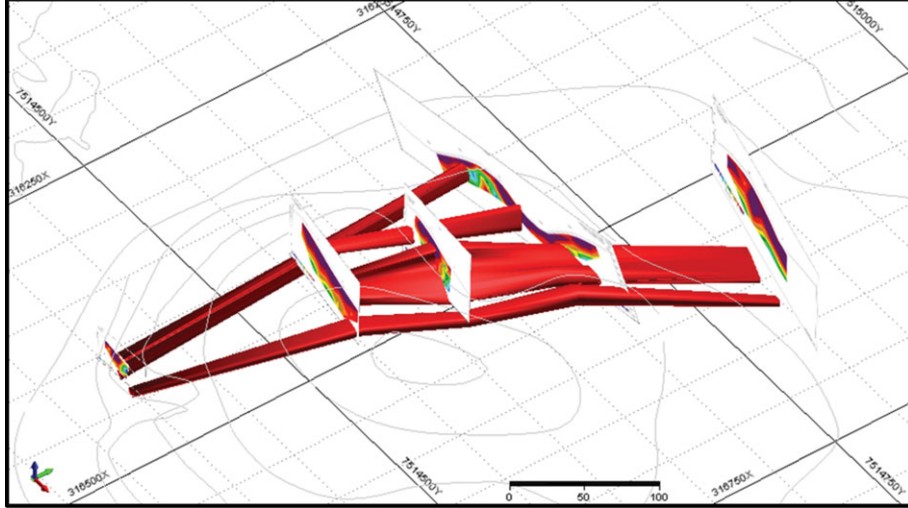

**Figure 16.** Preliminary model of the manganese deposit (modified from [24,25]).

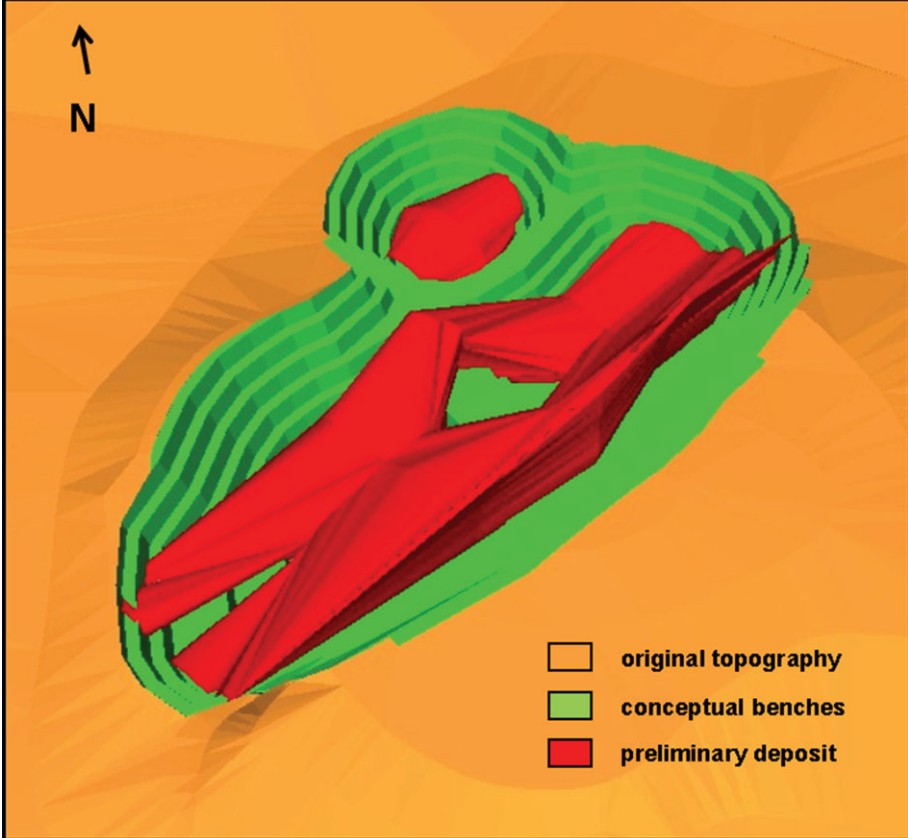

**Figure 17.** Conceptual pit design for the manganese deposit (modified from [34].

pit design as shown in figure 17. The designed conceptual pit was instrumental for tactical decision-making on how to plan mining operations at the manganese deposit in the medium-term horizon.

## 3.3. Short-term decision-making: limestone mine

The critical variables to be evaluated in the limestone deposit for short-term planning were the thickness of the soil, the thickness of the altered rock and the depth of the bedrock top. Geophysics VES surveys were selected as the method to identify the various layers investigated in this deposit. The initial step was

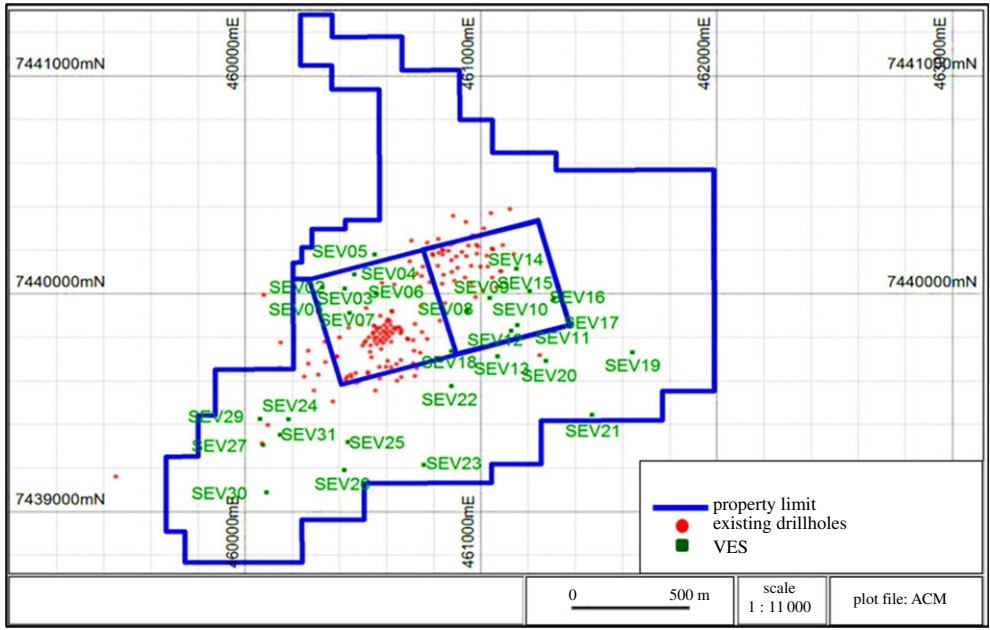

**Figure 18.** Location of vertical electrical sounding (VES) surveys (modified from [24,25]).

**Table 4.** Interpretation for VES10 and VES05 surveys (modified from [24,25]).

| VES | from (m) | to (m) | stratigraphy |
|---|---|---|---|
| 10 | 0.0 | 0.5 | soil |
| | 0.5 | 2.8 | altered rock |
| | 2.8 | 5.3 | altered rock—high moisture |
| | 5.3 | — | gneiss |
| 05 | 0.0 | 1.9 | Soil |
| | 1.9 | 8.5 | altered rock |
| | 8.5 | 13.8 | altered rock - high moisture |
| | 13.8 | — | gneiss |

the calibration of the VES-based resistivity and induced polarization methods against the core description of an existing drill hole in the same geographical location. The Schlumberger array, the most popular in VES applications, was chosen because it presents low sensitivity to lateral variations of resistivity and noises that exist underground, such as natural soil currents or interference from power lines. After calibration, the field team carried out 31 VES surveys in the southern portion of the mineral property, as shown in figure 18.

Most VES surveys have identified four layers, as observed in table 4. The first layer was interpreted as soil (low to medium resistivity), the second, unsaturated altered rock (high resistivity and medium chargeability), the third, saturated altered rock (low and medium resistivity and low chargeability) and the fourth, gneiss (high resistivity). Figure 19 shows the profile lines for geophysical interpretation and figure 20 displays the geological profiles resulting from interpretation of the VES information [24].

The goal of the VES survey carried out in the limestone mine was to incorporate a contiguous mineral property and to update the local geological model. However, because of the exploration project deadlines, the available time window for carrying out the exploration works in the entire contiguous mineral property was short and the activities had to be planned and executed within the mandatory deadlines. The geological profiles resulting from interpretation of the VES surveys were a critical tool for the operational decision-making on how to plan for mining at the new property in the short term.

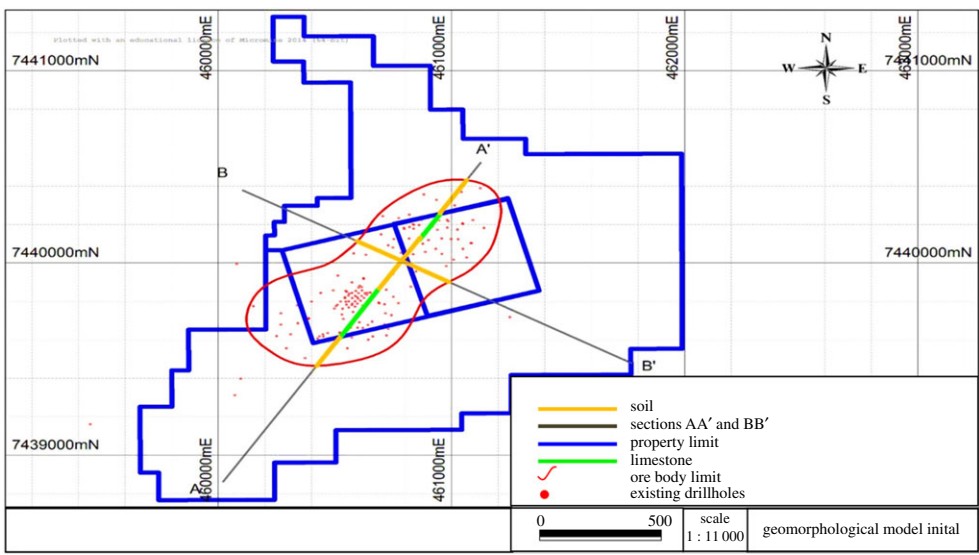

**Figure 19.** Profile lines for geological interpretation (modified from [24,25]).

# 4. Discussion

The use of VES and EP in the gold project, with the support of auger drilling assays, allowed for the construction of a geological model of the mineralized soil layer. This model supplied decision makers with information that reduced uncertainty and enabled the development of a conceptual pit design with 91% reduction in time and 70% cost savings as compared with a conventional, diamond-drilling survey. The conceptual pit helped decision makers to define a more realistic long-term mine plan for this project.

The execution of VES and EP works in the manganese deposit provided information that was incorporated in the geological model for the ore body. The model was employed in the development of a conceptual pit design for the estimation of key mining parameters such as the stripping ratio in a medium-term mining plan. The time reduction for mineral exploration was estimated as 77% and cost savings as 78% when compared with a conventional, diamond-drilling campaign.

The main result achieved with the VES campaign in the limestone mine was the construction of a geological model that determined the overburden volume in an adjacent mineral property. The updated geological model was used as the main tool for short-term mine planning in the new area. A time reduction of 75% was achieved and the exploration costs were reduced by 94% as compared with a conventional, diamond-drilling campaign.

Table 5 and figure 21 show the comparison of time and costs spent between the geophysical and diamond-drilling methods, including setting up and execution, for each of the projects analysed. Table 5 shows that the total cost of applying geophysics can be up to 94% lower than the total cost of a diamond-drilling campaign. For the limestone mine, 523 linear metres were surveyed with an estimated VES cost of R$228.50 for up to 50 m deep [36] and diamond-drilling in weathered rock at US$57.27 m$^{-1}$ [37].

The impact of geophysics on the reduction of geological uncertainty in small-scale mining in different mine planning horizons can be summarized by a graphic showing how the application of geophysical methods approximate the level of uncertainty of small-scale mining with the level of geological uncertainty in large-scale mining in long-, medium- and short-term mine planning.

For large-scale mining, data published by Ferreira [38] were adopted and the values for geological uncertainty in mine planning are 59% for long-term planning (LTP), 54% for medium-term planning (MTP) and 51% for short-term planning (STP). As for small-scale mining, the evaluation matrix was applied to the three mining projects before and after conducting the geophysical survey.

In the LTP horizon of the gold project, the initial index was 2.5, which corresponds to a level of geological uncertainty of between 75% and 95%. After the application of geophysics, the index turned out to be 3.2, or a level of uncertainty between 68% and 88%, equivalent to a reduction in geological uncertainty of approximately 8%.

In the MTP horizon of the manganese ore body, the index was originally 3.1, meaning that the geological uncertainty lies between 69% and 89%. With the application of geophysical methods, the

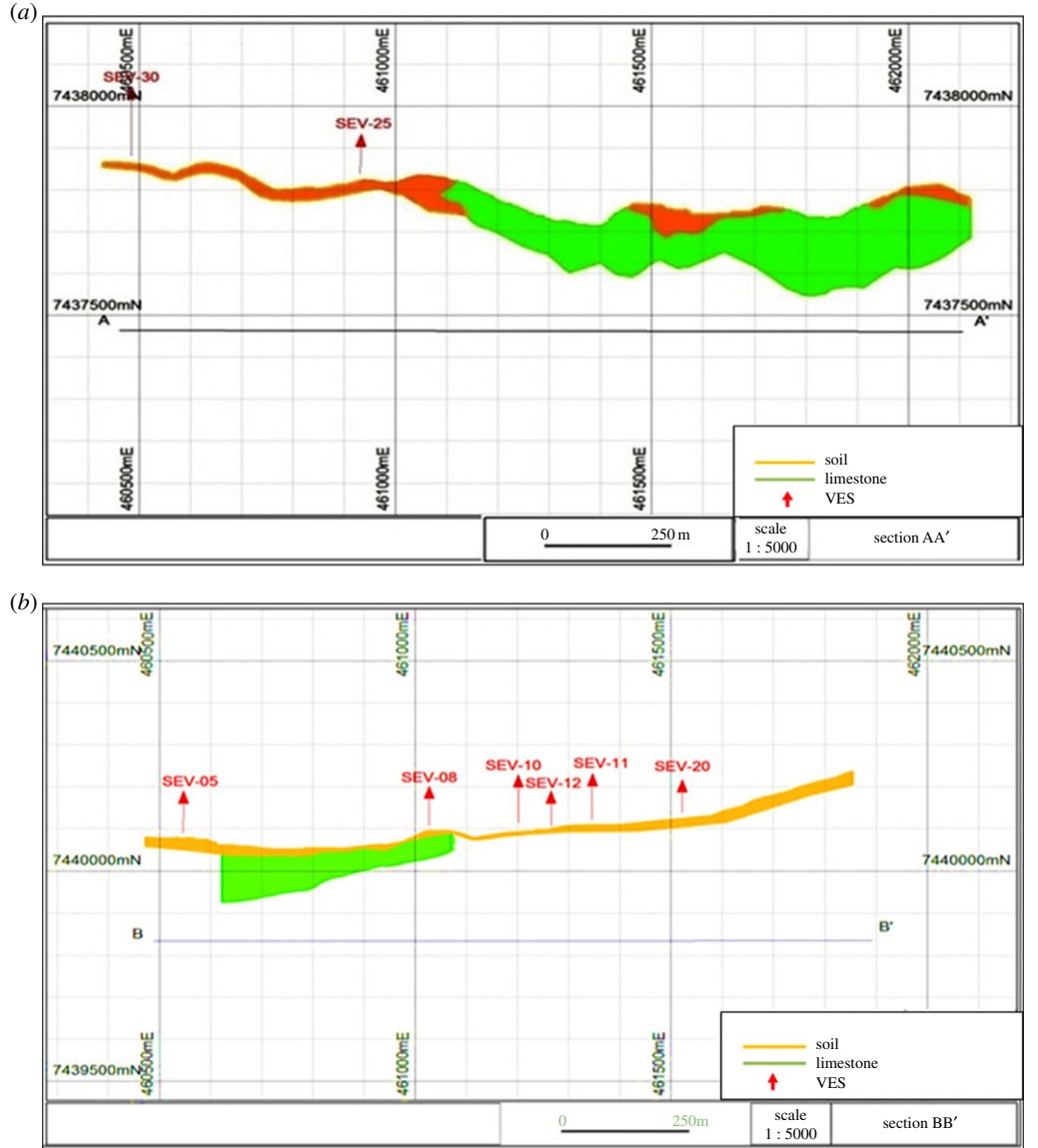

**Figure 20.** Geological profiles resulting from geophysical information (modified from [24,25]).

**Table 5.** Impact on time and costs of geophysics compared with drilling (LTP, long-term plan; MTP, medium-term plan; STP, short-term plan) (VES, vertical electrical sounding; EP, electrical profiling).

| project | techniques | time (days) | depth (m) | costs (US$/m) | total costs (US$) |
|---------|-----------|-------------|-----------|---------------|-------------------|
| gold (LTP) | geophysics (VES, EP) | 5 | 119 | 17.23 | 2.050 |
| | drilling campaing | 60 | 119 | 57.27 | 6.815 |
| manganese (MTP) | geophysics (VES, EP) | 7 | 172 | 12.52 | 7.520 |
| | drilling campaing | 30 | 172 | 57.27 | 34.394 |
| limestone (STP) | geophysics (VES) | 15 | 523 | 4.57 | 8.368 |
| | drilling campaing | 60 | 523 | 57.27 | 104.825 |

index increased to 3.9, which represents a level of uncertainty between 61% and 81%, a reduction of uncertainty of approximately 10%.

In the STP horizon of the limestone mine, the starting index was 4.0, which indicates a level of geological uncertainty between 60% and 80%. The data gathered with geophysics improves the index to 4.5, a level of uncertainty between 55% and 75% or a reduction of approximately 7%.

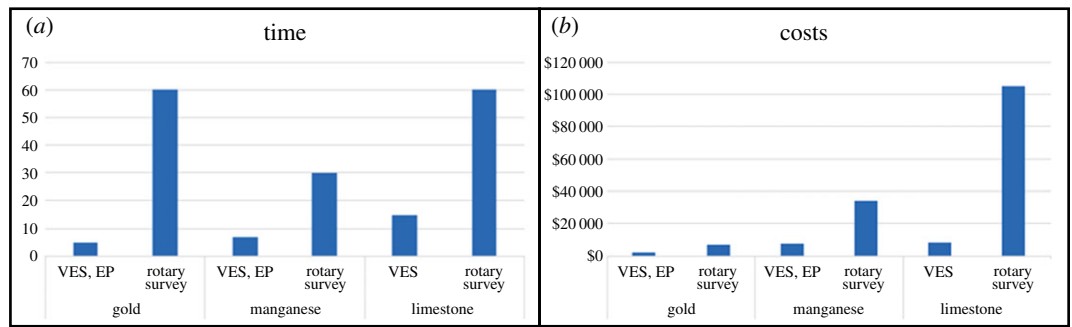

**Figure 21.** Comparison of time and costs spent between geophysics and rotary drilling (VES, vertical electrical sounding; EP, electrical profiling).

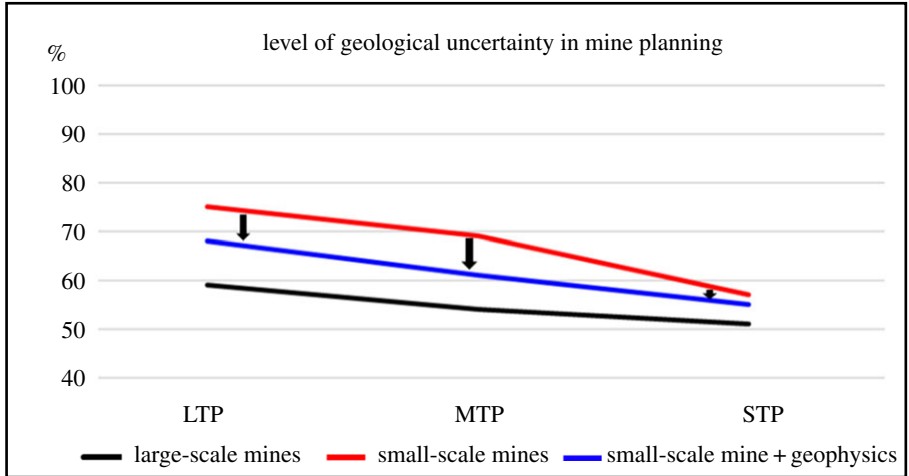

**Figure 22.** Level of geological uncertainty in large mines, small-scale mines and small-scale mines that apply geophysics (LTP, long-term plan; MTP, medium-term plan; STP, short-term plan) (modified from [35]).

Figure 22 shows the variation of geological uncertainty driven by the application of geophysical methods and techniques in small-scale mining, and how the deployment of geophysics approximate the small-scale mining line with the large-scale mining line.

## 5. Conclusion

The application of the proposed methodology has shown that combined geophysical methods can contribute to a reduction in the level of geological uncertainty in the strategic (long-term), tactical (medium-term) and operational (short-term) mine planning horizons in small-scale mining, when the appropriate conditions for obtaining reliable geophysics information are in place. The results indicated a reduction in geological uncertainty in long-term mine planning by approximately 6%, in medium-term planning by 13% and in short-term planning by 10%. This methodology allows for the construction of updated geological models, which are essential for proper decision-making in mine planning.

In addition, the application of geophysics delivered the necessary geological information in a shorter period and at a lower cost than traditional drilling programmes. This methodology saved 75% to 77% of mineral exploration time compared with conventional diamond-drilling campaigns. In the limestone mine, the exploration cost was reduced by 94% as compared with a diamond-drilling campaign.

The research has demonstrated that the application of adequate geophysical methods can contribute to the reduction of geological uncertainty in small deposits quickly at a low cost, while providing valuable information for geological modelling. As a result, more accurate modelling of mineral deposits allows decision makers to be more effective in strategical, tactical and operational mine planning, promoting higher productivity and improving the sustainability of small-scale mining projects.

Ethics. Ethical approval was obtained from the Centre for Responsible Mining of Universidade de São Paulo. The works executed and the data made available involved informed consent.

Data accessibility. The data from the surveys are freely available on Dryad at https://doi.org/10.5061/dryad.d7wm37pzh [39].

Authors' contributions. R.T. designed and wrote the manuscript. G.D.T. provided overall supervision for the research and reviewed the final text and dataset. A.C.M. was responsible for all field surveys, interpretations and results, and reviewed the project dataset. R.S.S. prepared all figures and tables and reviewed the final text and dataset. All authors gave final approval for publication.

Competing interests. We have no competing interests.

Funding. This study received financial support as detailed below: Coordenação de Aperfeiçoamento de Pessoal de Nível Superior - Brazil (CAPES) with grant nos. 88882.377505/2019-01 (R.T.), 1219401(A.C.M.) and FDTE 88882.377514/ 2019-011431.01.17 (R.S.S.), also supported by CNPQ (grant no. 3061177/2017-0).

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
