## [Reviewer comments · Royal Society Open Science]

Review History

RSOS-200384.R0 (Original submission)

Review form: Reviewer 1 (Antonis Vafidis)

Is the manuscript scientifically sound in its present form?

Yes

Are the interpretations and conclusions justified by the results?

Yes

Is the language acceptable?

No

Do you have any ethical concerns with this paper?

No

Have you any concerns about statistical analyses in this paper?

No

Recommendation?

Major revision is needed (please make suggestions in comments)

Comments to the Author(s)

This manuscript presents a novel application of a geophysical methodology on three case studies and demonstrates that this approach can reduce exploration cost in small scale mines. The original geophysical data from two sites (limestone and manganese mines) which were acquired using electrical resistivity methods (Vertical Electrical Sounding (VES) and the Electrical Resistivity Tomography (ERT) techniques), have been presented in another publication (Martins A.C., Terenci E.R., Tomi G.D., Tichauer R.M. 2016. Impact of Geophysics in Small-Scale Mining. J Remote Sensing & GIS, 5, 172). For these two case studies there is a significant overlap between the manuscript and the above mentioned paper regarding the text. The manuscript needs major revision to reduce this overlap.

The authors claim that they used the Induced Polarization (IP) method. There is no description of this method in "2.2 Geophysical methods". It is suggested to describe shortly the principle of IP method mentioning its usefulness in mining exploration or in delineating these particular deposits. Additionally, describe an IP survey from a single site (eg the limestone mine) including the experiment parameters, the employed instrument, the data processing techniques, the employed software and figures with selected resistivity and chargeability sections.

It is suggested to include figures with selected ERT section(s) and/or VES geoelectrical curves for the gold mine case study.

Some references are not properly written and/or are incomplete.

English needs improvement. More specific comments are attached in a file (Appendix A).

The manuscript needs major revision not only to reduce the overlap but also regarding the above mentioned comments.

Review form: Reviewer 2

Is the manuscript scientifically sound in its present form?

No

Are the interpretations and conclusions justified by the results?

No

Is the language acceptable?

Yes

Do you have any ethical concerns with this paper?

No

Have you any concerns about statistical analyses in this paper?

No

Recommendation?

Major revision is needed (please make suggestions in comments)

Comments to the Author(s)

The work in its present form is not suitable for publication and a major revision and not rejection is suggested in order to give an opportunity to authors to modify (if possible) their work.

The style of the work is suitable to a technical report or a Thesis and not to a scientific paper. The geophysical methods used/proposed are very old traditional geophysical methods and does not present any new methodological information nor an applied innovation.

In the abstract a "novel application" claimed. I can not see such in all over the text, even if the authors claim such in many parts of the text (see page 4, line 10-11).

If the approach presented in Martins et al (2016) and Martins (2017, Thesis) is new the authors have to present it within the text. This is missing .

Over all the work has no new methodology in its present form. The authors have to re-write the text making clear what is new and what is the real innovation claimed. In addition I strongly suggest to include in the revised text a clear contribution of how the geophysical uncertainty involved in the reduction of geological one. Furthermore, the references have to include some more recent ones and the existing part has to be updated with some references in the geophysical field from International mainstream journals. Till then the work is not suitable for publication.

Decision letter (RSOS-200384.R0)

Dear Mr Tichauer,

The editors assigned to your paper ("THE ROLE OF GEOPHYSICS IN ENHANCING MINE PLANNING DECISION-MAKING IN SMALL-SCALE MINING") have now received comments from reviewers. We would like you to revise your paper in accordance with the referee and Associate Editor suggestions which can be found below (not including confidential reports to the Editor). Please note this decision does not guarantee eventual acceptance.

Please submit a copy of your revised paper before 23-May-2020. Please note that the revision deadline will expire at 00.00am on this date. If we do not hear from you within this time then it will be assumed that the paper has been withdrawn. In exceptional circumstances, extensions may be possible if agreed with the Editorial Office in advance. We do not allow multiple rounds of revision so we urge you to make every effort to fully address all of the comments at this stage. If deemed necessary by the Editors, your manuscript will be sent back to one or more of the original reviewers for assessment. If the original reviewers are not available, we may invite new reviewers.

- Data accessibility

If you wish to submit your supporting data or code to Dryad (<http://datadryad.org/>), or modify your current submission to dryad, please use the following link:
<http://datadryad.org/submit?journalID=RSOS&manu=RSOS-200384>

- Competing interests

- Authors' contributions

- Acknowledgements

- Funding statement

on behalf of Professor Zach Agioutantis (Associate Editor) and R. Kerry Rowe (Subject Editor)

Comments to Author:

Reviewers' Comments to Author:

Reviewer: 1

Comments to the Author(s)

This manuscript presents a novel application of a geophysical methodology on three case studies and demonstrates that this approach can reduce exploration cost in small scale mines. The original geophysical data from two sites (limestone and manganese mines) which were acquired using electrical resistivity methods (Vertical Electrical Sounding (VES) and the Electrical Resistivity Tomography (ERT) techniques), have been presented in another publication (Martins A.C., Terenci E.R., Tomi G.D., Tichauer R.M. 2016. Impact of Geophysics in Small-Scale Mining. *J Remote Sensing & GIS*, 5, 172). For these two case studies there is a significant overlap between the manuscript and the above mentioned paper regarding the text. The manuscript needs major revision to reduce this overlap.

The authors claim that they used the Induced Polarization (IP) method. There is no description of this method in "2.2 Geophysical methods". It is suggested to describe shortly the principle of IP method mentioning its usefulness in mining exploration or in delineating these particular deposits. Additionally, describe an IP survey from a single site (eg the limestone mine) including the experiment parameters, the employed instrument, the data processing techniques, the employed software and figures with selected resistivity and chargeability sections.

It is suggested to include figures with selected ERT section(s) and/or VES geoelectrical curves for the gold mine case study.

Some references are not properly written and/or are incomplete.

English needs improvement. More specific comments are attached in a file (Specific comments.pdf)

The manuscript needs major revision not only to reduce the overlap but also regarding the above mentioned comments.

Reviewer: 2

Comments to the Author(s)

The work in its present form is not suitable for publication and a major revision and not rejection is suggested in order to give an opportunity to authors to modify (if possible) their work.

The style of the work is suitable to a technical report or a Thesis and not to a scientific paper. The geophysical methods used/proposed are very old traditional geophysical methods and does not present any new methodological information nor an applied innovation.

In the abstract a "novel application" claimed. I can not see such in all over the text, even if the authors claim such in many parts of the text (see page 4, line 10-11).

If the approach presented in Martins et al (2016) and Martins (2017, Thesis) is new the authors have to present it within the text. This is missing .

Over all the work has no new methodology in its present form. The authors have to re-write the text making clear what is new and what is the real innovation claimed. In addition I strongly suggest to include in the revised text a clear contribution of how the geophysical uncertainty involved in the reduction of geological one. Furthermore, the references have to include some more recent ones and the existing part has to be updated with some references in the geophysical field from International mainstream journals. Till then the work is not suitable for publication.

Author's Response to Decision Letter for (RSOS-200384.R0)

See Appendix B.

Decision letter (RSOS-200384.R1)

Dear Mr Tichauer,

It is a pleasure to accept your manuscript entitled "THE ROLE OF GEOPHYSICS IN ENHANCING MINE PLANNING DECISION-MAKING IN SMALL-SCALE MINING" in its current form for publication in Royal Society Open Science.

Please ensure that you send to the editorial office an editable version of your accepted manuscript - Word or LaTeX are preferred.

on behalf of Professor Zach Agioutantis (Associate Editor) and R. Kerry Rowe (Subject Editor)
openscience@royalsociety.org

Appendix A

Specific comments

Page	Line		Comment / Replace with
4	60	Haile (2014) comments that geophysical techniques are routinely used as part ...	Haile and Atsbaha , (2014) comments that geophysical techniques are routinely used as part ...
5		Frasheri (1995), the most important geophysical methods for mineral exploration are the ...	Frasheri et al (1995), the most important geophysical methods for mineral exploration are the
	19	... mining are based on the assessment conducted by Ferreira (2016), and for small-scale ...	Ferreira reference is missing
	21		Add more refs for geophysical apps in mining exploration
6	30	These methods are used to measure different electrical properties of rocks, such as electrical resistivity and magnetic permeability ...	These methods are used to measure different physical properties of rocks, such as electrical resistivity and magnetic permeability ...
	54	Electrical profiling is performed along imaginary lines in the terrain, resulting in a two dimensional profile for each line.	Electrical profiling is performed along survey lines in the terrain, resulting in a two dimensional profile for each line
7	21	Scores are integral numbers from 1 to 5 and measure the quality and broadness of implementation for each guideline.	Scores are integer numbers from 1 to 5 and measure the quality and broadness of implementation for each guideline.
8	30	The works were conducted in an area of 500 m ² , where there were outcrops of manganese-rich rocks, old mining works and agreement with landowners	The works were conducted in an area of 500 m ² , where there were outcrops of manganese-rich rocks and old mining works in agreement with landowners
9	6	One 120 m long electric profiling line spaced at 75 m;	Spaced from where?
	28	data obtained, through the VES and electrical profiling activities, indicate that a significant part of the manganese ore body can be found between 18.6 and 42.9 m deep and that the thickness can exceed 10 m.	Support this result ie. by showing it in a geophysical section where the values of a geophysical property (eg. resistivity) for the manganese ore body are different from the ones for the hosting formations.
	38	However, because of legal deadlines, the available time window for exploration works in the new mining area was short and the activities had to be planned and executed within the mandatory deadlines ..	However, because of exploration project deadlines, the available time window for exploration works in the new mining area was short and the activities had to be planned and executed within the mandatory deadlines ..

10	14	Most VES surveys showed four layers, as observed in Table 4. The first layer was interpreted as superficial soil (low to medium resistivity), the second, unsaturated altered rock (high resistivity and medium chargeability), the third, saturated altered rock (low and medium resistivity and low chargeability), and the fourth, gneiss (high resistivity).	Follow the same for the other two case studies. / Most VES surveys showed four layers, as observed on Table 4. The first layer was interpreted as superficial soil (low to medium resistivity), the second, unsaturated altered rock (high resistivity and medium chargeability), the third, saturated altered rock (low and medium resistivity and low chargeability), and the fourth, gneiss (high resistivity).
	28	The utilisation of VES and EP in the gold project, with the support of auger assays, allowed for the elaboration of a geological model for the mineralised soil layer ...	Can you include a figure with a resistivity and/or chargeability section in order to support this statement?
	40	The time reduction was calculated as 77% compared with a drilling campaign.	The time reduction for mineral exploration was calculated as 77% compared with a drilling campaign.
	46	A time reduction of 75% was achieved and the cost was reduced by 94% compared with a drilling campaign.	Can you add an example for one of these calculations?
11	10	For large-scale mining, data published by Ferreira were adopted and the values for geological uncertainty in mine planning are 59% for LTP, 54% for MTP and 51% for STP.	Ferreira reference is missing
	14	... before and after the conduction of geophysical assays..	before and after conducting the geophysical survey
	44	... that the application of combined geophysical methods..	... that combined geophysical methods..

The following Refs are not present in the text:

Braga A.C.O. 1997. Métodos geelétricos aplicados na caracterização geológica e geotécnica: formações Rio Claro e Corumbataí, no município de Rio Claro, SP. Instituto de Geociências e Ciências Exatas, UNESP, Rio Claro, 173.

Braga, A.C.O. 1999. Métodos de Prospecção em Hidrogeologia. Instituto de Geociências e Ciências Exatas, UNESP, Rio Claro.

Gamarra Chilmaza F. 2015. Programa para la prevención y eliminación progresiva del trabajo infantil en la minería artesanal de oro en sudamérica: estudio sociolaboral en los centros poblados de la Rinconada y Cerro Lunar. Puno: International Labour Organization.

Seccatore, J. 2014. An estimation of artisanal small-scale production of gold in the world. Science of the Total Environment, 496, 3–8.

Appendix B

To: Royal Society Open Science
Att: Professors Zach Agioutantis and R. Kerry Rowe
Date: 05/26/2020
Subject: Submission of Revised Manuscript

Dear Professors Zach Agioutantis and R. Kerry Rowe,

I am submitting for your appreciation the revised version of the manuscript "The Role of Geophysics in Enhancing Mine Planning Decision-Making in Small-scale Mining".

We wish to thank the Royal Society Open Science reviewers for the detailed suggestions and the valuable recommendations provided. The revision process has given us the opportunity to improve the quality of the manuscript significantly.

The main improvements are as follows:

- The style has been adjusted to a scientific paper format;
- The methodological innovation has been much clearer presented.
- The overlap with a published paper has been reduced to the minimum necessary;
- A brief explanation of the IP method has been inserted;
- A description of the geophysical parameters, equipment and processing techniques has been incorporated;
- ERT sections and VES geoelectrical curves have been added;
- References have been reviewed and incremented;
- English has been corrected.

Please find attached the review file with detailed explanations on the improvements made for each revision item. We are looking forward to receiving your feedback.

With my very best regards,

Ricardo Tichauer
MSc Mineral Engineering
PhD Mineral Engineering (ongoing)
USP - Universidade de São Paulo